# Early Detection of Diabetic Peripheral Neuropathy: A Focus on Small Nerve Fibres

**DOI:** 10.3390/diagnostics11020165

**Published:** 2021-01-24

**Authors:** Jamie Burgess, Bernhard Frank, Andrew Marshall, Rashaad S. Khalil, Georgios Ponirakis, Ioannis N. Petropoulos, Daniel J. Cuthbertson, Rayaz A. Malik, Uazman Alam

**Affiliations:** 1Diabetes & Endocrinology Research, Institute of Cardiovascular and Metabolic Medicine and The Pain Research Institute, University of Liverpool, Liverpool L69 7ZX, UK; hirkhal2@liverpool.ac.uk (R.S.K.); Dan.Cuthbertson@liverpool.ac.uk (D.J.C.); 2The Walton Centre, Department of Pain Medicine, Liverpool L9 7LJ, UK; Bernhard.Frank@thewaltoncentre.nhs.uk; 3Department of Musculoskeletal & Ageing Science, Faculty of Health & Life Sciences, Institute of Life Course & Medical Sciences, University of Liverpool, Liverpool L7 8TX, UK; Andrew.Marshall@liverpool.ac.uk; 4Faculty of Health and Life Sciences, The Pain Research Institute, University of Liverpool, Liverpool L9 7AL, UK; 5The Walton Centre, Department of Clinical Neurophysiology, Liverpool L9 7LJ, UK; 6Weill Cornell Medicine-Qatar, Qatar Foundation, Education City, Doha P.O. Box 24144, Qatar; g.ponirakis@gmail.com (G.P.); inp2002@qatar-med.cornell.edu (I.N.P.); ram2045@qatar-med.cornell.edu (R.A.M.); 7Institute of Cardiovascular Sciences, University of Manchester, Manchester M13 9PL, UK; 8Division of Endocrinology, Diabetes and Gastroenterology, University of Manchester, Manchester M13 9PT, UK

**Keywords:** diabetes, neuropathy, peripheral neuropathy, distal sensory polyneuropathy, diabetic neuropathy, diabetic peripheral neuropathy, early detection, screening, diagnostics, point-of-care

## Abstract

Diabetic peripheral neuropathy (DPN) is the most common complication of both type 1 and 2 diabetes. As a result, neuropathic pain, diabetic foot ulcers and lower-limb amputations impact drastically on quality of life, contributing to the individual, societal, financial and healthcare burden of diabetes. DPN is diagnosed at a late, often pre-ulcerative stage due to a lack of early systematic screening and the endorsement of monofilament testing which identifies advanced neuropathy only. Compared to the success of the diabetic eye and kidney screening programmes there is clearly an unmet need for an objective reliable biomarker for the detection of early DPN. This article critically appraises research and clinical methods for the diagnosis or screening of early DPN. In brief, functional measures are subjective and are difficult to implement due to technical complexity. Moreover, skin biopsy is invasive, expensive and lacks diagnostic laboratory capacity. Indeed, point-of-care nerve conduction tests are convenient and easy to implement however questions are raised regarding their suitability for use in screening due to the lack of small nerve fibre evaluation. Corneal confocal microscopy (CCM) is a rapid, non-invasive, and reproducible technique to quantify small nerve fibre damage and repair which can be conducted alongside retinopathy screening. CCM identifies early sub-clinical DPN, predicts the development and allows staging of DPN severity. Automated quantification of CCM with AI has enabled enhanced unbiased quantification of small nerve fibres and potentially early diagnosis of DPN. Improved screening tools will prevent and reduce the burden of foot ulceration and amputations with the primary aim of reducing the prevalence of this common microvascular complication.

## 1. Introduction

The International Diabetes Federation (IDF) estimated the global prevalence of diabetes is 425 million people in 2017 and is predicted to rise to 628 million by 2045 [1]. This has been accompanied by an increase in the burden of diabetic complications [2,3]. Diabetic neuropathy affects 10–50% of people with type 1 (T1D) and type 2 diabetes mellitus (T2D) [4,5,6,7]. In the US, the annual cost for managing DPN and foot ulceration with lower limb amputation is estimated to be between $4.6–13.7 billion [8]. Diabetic peripheral neuropathy (DPN) has a predilection for small unmyelinated or thinly myelinated C and Aδ nerve fibres [9], which mediate temperature and pain perception, tissue blood flow and sweating, all of which are key factors for foot ulceration [10]. Small fibre deficits are considered to precede large fibre involvement in DPN [5,11]. Furthermore, small fibre degeneration occurs in prediabetes suggesting early subclinical pathology before the onset of overt T2D [12,13]. Indeed, small fibres are the earliest to degenerate and have the greatest potential for repair as shown in studies with normalisation of hyperglycaemia through pancreatic transplantation in T1D and weight loss following lifestyle intervention in prediabetes [14,15,16].

### 1.1. Economic and Functional Consequences of Small Fibre Degeneration

Degeneration of small sensory nerve fibres occurs in painful DPN (pDPN) which is present in up to one-third of patients with diabetes [17,18,19]. Neuropathic pain has a profound impact on quality of life, physical and emotional health and affects both functionality and sleep [20,21,22]. Chronic intractable pain is associated with anxiety and depression and is often refractory to current therapies [21,23]. Consequently, people with pDPN are more likely to be unemployed and loss of working time in a U.S. population cost ~$3.65 billion each year [24,25]. Furthermore, people suffering severe chronic pain have an increased ten-year mortality [26].

DPN is significantly underdiagnosed leading to missed opportunities for preventing progression to severe DPN and foot ulceration, which has a dreadful 5-year mortality [27,28,29]. Indeed, DPN is a major cause of foot ulceration and is implicated in 50–75% of all non-traumatic amputations [30,31]. Mortality, one and five-years after lower limb amputation in people with diabetes ranges from 10–50%, to 30–80% respectively [32,33,34] with the latter mortality rate comparable to lung cancer [35]. DPN and amputation represent a devastating impact on the individual leading to a loss of function, quality of life and financial stability [36,37]. In the UK, the National Health Service (NHS) spent £639 million on diabetic foot ulcers and £662 million on lower limb amputations, accounting for £1 in every £150 spent out of the NHS healthcare budget [38].

### 1.2. Pathogenesis

Whilst there has been progress in identifying the pathophysiology of DPN, a complete understanding of this process remains elusive [39]. DPN is associated with hyperglycaemia, hyperlipidaemia, insulin resistance and protein catabolism [6,40]. Hyperglycaemia-induced oxidative stress and reactive oxygen species result in peripheral nerve injury [41,42]. Experimental data have demonstrated nitro-oxidative stress in dorsal root ganglia, axons and Schwann cells with nerve conduction impairment, neurovascular dysfunction, apoptosis and sensory deficits [43,44,45,46]. There is also activation of poly (ADP-ribose) polymerase, polyol, hexosamine and protein kinase C (PKC) pathways and accumulation of advanced glycation end products culminating in axonal dysfunction and damage [44,47,48,49,50,51]. Increased flux through the polyol pathway leads to accumulation of sorbitol and fructose, myo-inositol depletion and a reduction in Na^+^K^+^-ATPase activity. Endoneurial microvascular deficits result in hypoxia and ischaemia, generation of reactive oxygen species (oxidative stress), activation of the redox-sensitive transcription factor NFκB, and increased activity of PKC [52,53].

### 1.3. Evidence in Favour of Early Intervention for DPN

The incidence of DPN is associated with hyperglycaemia and also cardiovascular risk factors such as raised cholesterol, triglycerides, hypertension, obesity and smoking [54]. Indeed, all of these risk factors can be modulated by early intervention. In a large longitudinal study of 1441 people with T1D in the Diabetes Control and Complications trial (DCCT), intensive insulin treatment reduced the risk of developing DPN by 60% [55,56]. In fact, a continuous beneficial effect after intensive insulin treatment was observed in participants of the DCCT trial after 10 years of follow-up in the Epidemiology of Diabetes Interventions and Complications (EDIC) trial [57]. Furthermore, a Cochrane Systematic Review found enhanced glucose control significantly reduced the risk of developing DPN in participants with T1D compared to standard of care [58]. However, whilst tight glucose control reduces the incidence of DPN in T2D, this reduced risk was not statistically significant [58] and in T2D, DPN has greater multifactorial causality due to the heterogeneous nature of the disease. For instance, obesity and hypertriglyceridemia are significant risk factors of DPN for people with T2D, independent of glucose control [59]. It follows that treatment of hypertension in people with T2D is associated with a significant reduction in the incidence of DPN and improvements in people with mild DPN [60,61,62]. Furthermore, a small randomised, double-blind, placebo-controlled Phase IIa study of participants with T2D and early DPN, found that reduction of low-density lipoprotein (LDL) cholesterol and triglycerides using rosuvastatin improved the neuropathy score and nerve conduction parameters [63]. Individualised diet and aerobic and resistance exercise regimens are important in the reversal of early DPN changes and the prevention of progression to DPN [15,64,65]. Thus, a multifactorial approach is required for the prevention and early treatment of DPN of people with T2D. A key underpinning of multifactorial treatment is an accurate, reiterative diagnostic modality for the screening of people with diabetes to reliably detect early DPN.

### 1.4. Current Clinical Assessment of Neuropathy

The signs and symptoms of DPN are insidious and current screening programmes rely on subjective tests of large nerve fibre dysfunction [66,67]. NICE recommends vibration perception testing using a 128 Hz tuning fork together with a 10 g (Semmes-Weinstein) monofilament for the screening of DPN [67]. However, these tests identify DPN at a late, irreversible, pre-ulcerative stage [68,69,70]. Thus, an abnormal monofilament test is associated with a 3-year relative risk of 15% (95% CI 9.0 to 26.0) for foot ulceration or lower limb amputation [71]. Despite, early and progressive injury to small fibres in diabetes, small nerve fibre assessment is not included in annual diabetic foot screening programmes [11,72].

In direct contrast, diabetic retinopathy and diabetic kidney disease have effective screening programmes which detect early sub-clinical pathology, enabling early interventions [73] which has led to a reduction in blindness [73] and end stage renal failure [74]. In fact, largely due to the success of diabetic retinopathy screening, it is no longer the leading cause of sight loss in western society [75]. Early, multifactorial risk factor modification may reduce the risk of foot ulceration and amputation [76,77]. The Toronto Diabetic Neuropathy Expert Group and American Diabetes Association (ADA) [6,78] have recommended the early detection and monitoring of DPN, but have recommended monofilament or crude neurological testing. Clearly, there is a need for robust screening methods capable of diagnosing subclinical DPN. This article aims to critically appraise commonly used research and clinical diagnostic tools to evaluate their potential role in screening for early DPN.

### 1.5. Methods

Electronic database searches were undertaken in Google Scholar, EMBASE, PubMed, OVID and Cochrane CENTRAL to identify the included articles. Reference lists of relevant articles were searched and in addition, studies were identified by authors with expertise in DPN. Studies published from initial curation of the electronic database to November 2020 were identified and those felt not relevant by authors were excluded with the guidance of the senior author (U.A.).

## 2. Screening and Diagnostic Tools

### 2.1. Composite Scoring Systems

Confirmed DPN represents a subtle and gradual disease process, in which the symptoms are often unreliable as an indicator of early nerve damage [79,80]. However, neuropathic pain may be the initial presenting symptom of diabetic neuropathy in patients with diabetes, pre-diabetes or metabolic syndrome [8]. Thus, validated screening instruments which utilise sensory and affective verbal pain descriptors (burning sensation, tingling/prickling, numbness, electric shocks, pain evoked by light touch) such as the Leeds Assessment of Neuropathic Symptoms and Signs (LANSS), douleur neuropathique en 4 (DN4) and painDETECT are widely used for the identification of neuropathic pain in diabetes [9,10,11,12,13,14]. Notably, Bennet et al. [15] demonstrated the utility of screening for neuropathic pain in a population of people with diabetes using a postal self-completed portion of the LANSS questionnaire, highlighting the excess prevalence and burden of pDPN. However, measures such as the neuropathy symptom score (NSS) cannot reliably identify early DPN [81,82]. Numerous clinical scoring systems have been compiled to evaluate light touch, pin-prick, vibration, proprioception, muscle strength and ankle reflexes [83]. The Michigan neuropathy screening instrument (MNSI) evaluates both positive and negative sensory symptoms and an examination of the foot to identify dry skin and ulcers [84]. In a large cohort of 1100 people with T1D Herman et al. [85] showed that the MNSI questionnaire had a low sensitivity, high specificity and only a moderate negative predictive value (NPV) compared to nerve conduction studies (NCS). Furthermore, the MNSI questionnaire was unable to identify the development of DPN over 13 years in a cohort of 1256 participants with T2D in the ACCORD-Denmark study [86]. VPT and NCS identified a higher prevalence of DPN compared to the MNSI and physical examination in a cohort of participants with T2D [87].

The neuropathy disability score (NDS) compiled by Dyck et al. [81] is a composite scoring system that assesses signs of neuropathy. A simplified NDS has been utilised widely to identify the signs and severity of DPN using a 0–10 scoring system [88], with a score of ≥6 to define established neuropathy [89] and can also be used to stratify patients into mild (3–4), moderate (5–6) or severe neuropathy (7–10) or those at high risk of foot ulceration as it is weighted for large fibre testing [90].

The neuropathy impairment score (NIS) has been adapted to assess DPN by focusing the examination to the lower limbs (NIS-(LL)) incorporating an assessment of muscle weakness, reflexes and sensory loss [91]. The NIS-LL+7 includes the NIS and seven additional tests including five different NCS of the lower limbs, vibration detection and heart rate response to deep breathing. NIS-LL+7 has demonstrated the capacity to identify early changes in DPN and worsening in a longitudinal study [82]. However, the inclusion of multiple tests into a single composite score is labour intensive and time consuming and is not suitable for screening [92]. These composite scoring systems and the current annual diabetic foot check are not effective in the detection of early DPN.

### 2.2. Thermal and Vibration Perception Thresholds

The detection of sub-clinical neuropathy in both symptomatic and asymptomatic patients with diabetes mellitus in the early stages of the disease is key to providing a window of opportunity for optimising multifactorial treatment and limit the progression of DPN [93]. The routine use of thermal thresholds in clinical practice has been difficult to implement due to the cost and subjective nature of the testing and a lack of consensus on standard practice [94,95]. Abnormal thermal sensation has been identified in over 93% of subjects with impaired glucose tolerance (IGT) or T2D [96] and in ~50% of asymptomatic participants with diabetes and normal NCS [97].

A large cross-sectional study found participants with T2D had a higher heat detection threshold (HDT) and lower cold detection threshold (CDT) of both the feet and hands compared to healthy volunteer control participants [98]. Moreover, in participants with T2D the prevalence of abnormal thermal thresholds in the big toe (60.2%) and on the dorsum of the foot (45.2%) was higher than abnormalities in sural nerve action potential (SNAP) (12.9%) [98]. Thermal threshold abnormalities are also present in participants with early-onset T2D with further abnormalities in painful and painless DPN compared to controls [99,100,101]. Interestingly, thermal threshold testing outperformed intra-epidermal nerve fibre density (IENFD) in the detection of an abnormality (abnormal thermal thresholds: 50% vs. reduced IENFD: 40%) in a cohort of 210 participants with signs and symptoms of neuropathy [102]. Detecting sensory abnormalities using VPT, pinprick, thermal thresholds and light touch testing in participants with confirmed neuropathy has demonstrated high specificity (77–96%) and positive predictive value (PPV) (95–98%) for DPN [103]. Thus, thermal thresholds and pinprick testing may be more useful in identifying early DPN as they test C and Aδ fibres whilst VPT and light touch are measures of large Aβ fibre function [104,105,106,107]. The results of VPT, pinprick and light touch testing modalities using a standardised protocol have demonstrated good reliability in healthy controls and people with DM [108,109,110]. However, a wide range of variability has been reported across studies ranging from poor to excellent for cold detection threshold (CDT), heat detection threshold (HDT), cold pain threshold (CPT) and heat pain threshold (HPT) [111,112].

### 2.3. German Network on Neuropathic Pain

The protocol developed by the German Research Network for Neuropathic Pain (DFNS) is comprised of the following components; mechanical detection threshold (MDT), mechanical pain threshold (MPT), C-tactile afferents [113], wind-up ratio (WUR) [114], pressure pain threshold (PPT), vibration detection threshold (VDT), thermal thresholds (TT) and thermal sensory limen (TSL) [115]. The loss of small nerve fibre sensitivity or gain of function can be detected using QST for different nerve fibre populations [116,117]. The DNFS have published age and sex matched normative data for discrete areas of the trunk, face, hand and foot for participants aged 6–75 years [118,119,120]. The DNFS QST static protocol differs from bed-side sensory testing through standardisation of the stimuli and standardised instructions to the patient and examiners to execute the static protocol [121,122]. Dynamic QST assesses the change in sensitivity of a test before and after a painful stimulus to identify mechanisms of pain processing as opposed to the static QST limen which measure the basal states of the nociceptive system [115]. Whilst no reference data are currently available for dynamic QST, practical recommendations have been published [123].

Notably, Kopf et al. [124] identified sensory abnormalities in 71% of participants with pre-diabetes, and 95% of participants with T2D, outperforming NCS. Further, participants with T2D had greater deficits in large fibre function and CDT with a gain of function in both MPT and PPT compared to pre-diabetes [124]. Correspondingly, a large study of a cohort of 350 participants with polyneuropathy found that participants with small fibre neuropathy had a loss of Aδ and C-fibre function (CDT, WDT and TSL; all *p* < 0.001) whilst participants with polyneuropathy had loss of function in both small and large fibres (CDT, WDT and TSL; all *p* < 0.001, MDT and VPT; both *p* < 0.001) compared to 273 controls [125].

Maier et al. [110] showed the suitability of QST for identifying sensory abnormalities in 1200 participants with a wide range of polyneuropathies as there was at least one sensory abnormality in 92% of all patients with neuropathy. Interestingly, IENFD can be significantly decreased despite, normal sensory detection thresholds [126]. In a recent criticism, Schmelz [127] argues that QST is unable to differentiate painful from non-painful neuropathies [128]. Importantly, the German DNFS QST protocol does not directly differentiate between central nervous system or peripheral nerve involvement [129], but may be used as a supplementary diagnostic modality characterising sensory phenotypes [122,130,131]. Thus, the combination of QST together with another functional or structural measure of neuropathy such as NCS, IENFD or corneal confocal microscopy (CCM) is suggested [12,132]. QST may be useful to identify patients who may have small fibre deficits, especially asymptomatic patients with normal NCS [133,134]. A large multi-centre study found low heterogeneity across the full static DNFS QST protocol [135]. Environmental and methodological factors and the capacity of the participant to interact appropriately with the QST testing protocol are pivotal to the quality assurance of QST [112,122], but is reliable and reproducible in generating sensory phenotypes.

### 2.4. Evoked Potentials

The European Federation of the Neurological Societies (EFNS) have identified evoked potentials as a non-invasive, reliable tool for investigating Aδ fibre function in patients with neuropathic pain [131]. Laser evoked potentials (LEPs) have been identified as “grade A” evidence by the EFNS [131]. The response of primary nociceptive afferents to LEPs and contact heat evoked potentials (CHEPs) assesses the activation of the primary and secondary somatosensory cortex, insula and mid-cingulate cortex [136,137]. LEPs are elicited by skin stimulation with afferent radiant heat emitted by a CO_2_ or solid-state laser [138,139]. A𝛿-fibre function is recorded through scalp electrodes measured at a late range latency of the brain action potential (200–400 ms) [116]. Age-corrected normative reference ranges from LEPs have been published from healthy controls and validated against IENFD [140]. Although normative data has been published, no consensus has been reached on CHEP methodology, cautioning against the use of these normative values if the methodology is different for eliciting CHEPs [141,142,143,144]. Further, CHEP amplitudes decrease with age in healthy control subjects and vary between male and female participants [142,143,145].

Differences between participants with polyneuropathy and controls have been identified using CHEPS [146,147,148]. Reduced CHEPS amplitudes (prolonged N2 latencies) were identified in 61.2% of patients with symptoms of a length dependent somatosensory neuropathy detecting a greater prevalence of abnormalities than thermal thresholds and NCS [141,149]. In a cross-sectional cohort study, both LEPs and CHEPs latency were reduced in participants with mixed-fibre polyneuropathy, but in participants with small fibre pathology only CHEPS were reduced compared to controls [150].

A small cohort study by Ragé et al. [151] identified a significant reduction in LEP amplitude in participants with T2D compared to T1D (*p* = 0.022) and healthy controls (*p* = 0.027). Similarly, Di Stefano et al. [140] found that LEPs were significantly reduced in participants with DM and hyperalgesia compared to participants with hypoalgesia (*p* < 0.05) [140]. Further, participants with reduced LEPs had a significantly reduced IENFD. Notably, reduced LEPs in the lower limb have been identified in people with diabetes and small fibre neuropathy [133,151]. In a large cross-sectional study by Wu et al. [141] 188 participants with neuropathy had significantly reduced CHEP amplitudes and longer N2 latencies compared to controls (21.7 ± 14.8 vs. 33.8 ± 12.1 µV, *p* < 0.001; 527.4 ± 7.4 vs. 495.1 ± 41.4 milliseconds, *p* < 0.001, respectively). CHEPS has a greater diagnostic efficiency compared to HDT thresholds on the dorsum of the foot (*p* = 0.036) [141]. CHEPS shows good sensitivity (78%), specificity (81%), PPV (84%) and NPV (75%) of LEPs based on a disease threshold of age-adjusted normative mean values, using IENFD as the reference standard [140]. The diagnostic efficiency of evoked potentials is comparable to IENFD (Table A1) but they are non-invasive [140].

However, similar to QST, evoked potentials cannot delineate peripheral versus central nociceptive pathway pathology [129]. The limited availability of equipment and specialist nature of evoked potential evaluation limits this test to specialist neurological centres.

### 2.5. Microneurography

Microneurography detects compound action potentials of peripheral nerves that can be recorded using a Tungsten needle electrode percutaneously inserted in a nerve fascicle [136,152,153]. The electrode can be carefully manipulated to record multiple or singular action potentials from large myelinated fibres, unmyelinated fibres, mechano-sensitive C-fibres and heat-sensitive C-fibres [136,154,155,156,157]. Microelectrodes are typically inserted superficially to assess the cutaneous branches of the peroneal nerve at the fibular head [156]. The skin is stimulated with needle electrodes to identify C-fibres with comparatively slower conduction velocities of <2 ms^−1^ [153]. Thus, the action potentials during electrical stimulation are visualised in real time using a spike raster plot [157].

Microneurography has demonstrated abnormal sensitivity to thermal and mechanical stimuli which evoked a doubling of C-fibre action potentials and spiking in patients with either symptoms of neuropathy or confirmed painful neuropathy [154,155]. Further, spontaneous activity of hyper-excitable nociceptive C-fibres was identified in 87% of participants with pDPN [158]. Thus, sensitisation of C-fibres is associated with hyperalgesia in painful neuropathy [159,160]. Moreover, there are more hyper-excitable sensitised mechano-insensitive C-fibres in participants with painful compared to painless neuropathy [156]. Ørstavik et al. [157] identified a higher proportion (2:1 ratio) of mechano-insensitive to afferent mechano-responsive C-fibres in patients with pDPN or without DPN compared to controls (1:2 ratio).

However, microneurography is invasive and requires highly skilled operators. Furthermore, the protocols described for a complete recording of C-fibre function can approach 7 h per session [156] and is therefore primarily used in a research setting.

### 2.6. Current Perception Threshold

An additional modality used to measure current perception threshold is an electro-diagnostic device which assesses the functionality of Aβ, Aδ, and C type fibres by measuring current perception threshold at 2000, 250, and 5 Hz respectively [161]. Current evidence suggests it is a useful, non-invasive technique to evaluate DPN in the early, asymptomatic stages [161,162]. One study involving 558 participants with T2D, indicated that using a neurometer to measure current perception threshold identifies a greater number of DPN cases compared to monofilament testing (current perception threshold 33.9% vs. monofilament 10.6%) [163]. Similarly, a retrospective study involving 202 participants with T2D, found a greater number of subclinical DPN cases using current perception threshold compared VPT [164]. In another study involving 52 participants with T1D, observed that current perception threshold of the bilateral median and sural nerves was significantly lower in participants with diabetes [165]. Normative data from 166 healthy participants found measurements of the hand, finger and big toe are influenced by both age and sex [166]. Further studies are warranted to identify a disease cut-off value together with studies to indicate the sensitivity and specificity of current perception threshold for the screening or diagnosis of DPN.

### 2.7. NC-Stat DPN Check

The NC-Stat DPN Check is an FDA approved point-of-care nerve conduction device [167,168] that evaluates sural nerve function at the lateral malleolus [167]. Whilst there is a correlation between SNAP, SNCV and measures of small fibre function, the NC-Stat DPN Check is a measure of large fibre function and does not directly give information on small fibres [169]. In a recent study [170], there was a good correlation between standardised NCS and NC-Stat DPN Check with SNCV (*r* = 0.81) and a moderate correlation for SNAP (*r* = 0.62) [170]. The NC-Stat DPN Check has high reproducibility and can offer an objective measure of large fibre function. A intra-class correlation coefficient of 0.97 for SNAP and 0.94 for SNCV has been demonstrated with NC-Stat DPN-Check [171]. Inter-observer reproducibility compared favourably with reference NCS with correlation coefficients of 0.83 for SNAP and 0.79 for SNCV [171]. However, DPN-Check has been reported to over-estimate SNCV compared to reference NCS by a mean of +8.4 ± 6.4 m/s, representing consistent over-estimation bias [171].

In a recent cohort study of T2D participants and age and sex matched controls the diagnostic accuracy of NC-Stat DPN Check for either SNAP (<4 µV) or SNCV (<40 m/s) in one or both legs was compared with the NDS (≥3) as a clinical measure of DPN [172]. Compared to the NDS, the NC-Stat DPN Check had a sensitivity of 90.4%, specificity of 86.1%, PPV of 79.1%, NPV of 93.9%, positive likelihood ratio (LR+) of 6.51 and negative likelihood ratio (LR−) of 0.11 [172]. Smaller studies have shown that NC-Stat DPN Check has a sensitivity of 80–92% and a specificity of 80–82% for DPN [173,174,175]. The use of different disease thresholds for SNAP and SNCV in older participants and those with long-standing diabetes affects the diagnostic efficiency of NC-Stat DPN Check (Table A2). Sharma et al. [176] suggest that NC-Stat DPN Check could be used to triage and identify abnormal NCV before undertaking confirmatory NCS using standard electrophysiology equipment. However, NC-Stat DPN Check comes with a cost and any subsequent reference NCS requires specialist clinics and highly trained staff.

### 2.8. Skin Biopsy

Skin biopsy of the distal leg and thigh with quantification of intra-epidermal nerve fibre density (IENFD) is considered to be the “gold standard” to diagnose small fibre neuropathy [6,177]. The thigh and lower leg are considered to be the optimal sites for biopsy [178] to identify reduced IENFD in subjects with length-dependent neuropathy [179]. Normative age-matched data are available for IENFD [180,181]. The methods for processing skin biopsy samples and quantifying small nerve fibre pathology have been standardised and are detailed in the ENFS guidelines [177]. Specialist facilities and experience are required to produce reliable IENFD staining with PGP9.5 which is time consuming and expensive. IENFD is demonstrated using the pan-axonal marker protein gene product 9.5 (PGP-9.5), or more specifically the retrieved antigen ubiquitin carboxyl terminal hydrolase using immunohistochemistry or immunofluorescence [177,182,183,184]. IENFD has a sensitivity of 61–97% and specificity of 64–95% for identifying small fibre pathology (Table A1) [101,102,133,185,186,187,188,189,190]. IENFD cannot currently discriminate between pDPN and painless DPN [191,192]. However, deficits in regenerative capacity due to neurovascular dysfunction and inflammation may cause neuronal injury to outpace repair [46,193]. Thus, the regenerative capacity of intra-epidermal nerve fibres may help to discriminate pDPN from DPN by staining with the neuronal regeneration marker growth associated protein-43 (GAP-43) and assessing the ratio of GAP-43^+^ to PGP-9.5^+^ nerve fibres [99,194,195]. Due to contradictory published data further work is required to validate this approach [195]. IENFD was assessed in patients with T2D over 5 years and demonstrated that nerve regeneration was overtaken by neurodegeneration [196]. The rate of intra-epidermal nerve fibre loss in patients with DPN has been identified as 3.76 ± 1.46 fibres/mm per year. Although skin biopsy is minimally invasive [177], there is risk of bleeding and infection which makes this method less appealing as a screening method for DPN [197].

### 2.9. Sudoscan

Sudoscan provides a quantitative measurement of sudomotor function by quantifying electrochemical skin conductance (ESC) of the hands and feet as a measure of postganglionic sympathetic integrity [151,198]. Indeed, the EZ Scan device is an FDA approved device to identify and risk-stratify subjects with pre-diabetes or undiagnosed DM [199,200,201,202]. An abnormality in the EZ scan has been associated with progression of diabetic retinopathy and autonomic neuropathy [203,204]. Sudoscan on the other hand, is an FDA approved point of care device [205] which has been advocated as a screening tool for DPN [206]. It is quick (3–5 min) and does not require trained personnel or specialised facilities [207]. The change in conductance is calculated after stimulation of the skin by a low-voltage current (≤4 volts) through reverse iontophoresis or chronoamperometry of chloride ions [207]. Sudomotor dysfunction is more prevalent in the feet compared to the hands in people with diabetes [208].

In a study of 394 subjects with T2D, lower ESC in the feet was associated with higher NSS, NDS and VPT [209]. Notably, abnormalities in ESC have been identified in 69% of participants with asymptomatic DPN [209]. In a large cross-sectional study of 523 participants with T2D, Sudoscan was more sensitive for the detection of DPN compared to NDS and VPT with a sensitivity and specificity of 85% (AUROC = 0.88) [210]. Notably, Sudoscan in a mixed cohort of patients with distal polyneuropathy (*n* = 55; 22 with diabetes, 2 prediabetes, 31 idiopathic) and controls (*n* = 42) yielded a sensitivity of 77%, specificity 67%, PPV 59% and an NPV of 83% which was comparable to the diagnostic efficiency of IENFD of the lower leg (AUC 0.76 and 0.75 respectively) [211]. The diagnostic ability of Sudoscan in smaller cohort studies in participants with T2D are summarised in Table A3.

It has been suggested that Sudoscan may be used as an initial screening tool for DPN. However, Rajan et al. [212] have criticised the body of evidence supporting Sudoscan due to the heterogeneous normative values across different populations.

### 2.10. Neuropad

The Neuropad test kit contains two adhesive plasters containing anhydrous blue salt cobalt-II-chloride, which changes colour from blue to pink upon exposure to sweat [213] and the colour change after 10 min has been used to identify the severity of sudomotor dysfunction [214]. Neuropad allows objective assessment of sudomotor dysfunction, particularly in older patients who may lack the capacity to engage with the standard-of-care tests [215]. The simplicity and ease of interpretation of the results allows for self-assessment to identify sub-clinical DPN [216]. The time to complete colour change correlates with the MNSI [217], which correlates with DPN severity [218] and diabetes duration [219]. A significant reduction in IENFD has been demonstrated in people with diabetes and abnormal or patchy Neuropad results [220]. In a large, multi-centre, cross-sectional study 1010 participants with T2D underwent NDS as a reference test to identify DPN [219]. An abnormal Neuropad response in one or both legs was associated with a 94.9% sensitivity, 70.2% specificity, 46.3% PPV and 98.1% NPV for DPN compared with a NDS >6 [219]. Multiple cross-sectional studies have reported a high sensitivity and moderate specificity for the detection of sudomotor dysfunction (Table A4). Hewitt et al. [221] report a sensitivity of 89.4% (95% CI 83.2–93.5) and a specificity of 60.3% (95% CI 50.9–69) for the diagnosis of DPN. Moreover, sudomotor dysfunction and DPN was identified using Neuropad in 43.4% patients with recently diagnosed DM (*n* = 151) [222]. Despite the higher sensitivity the lower specificity of Neuropad compared to the 10 g monofilament [223] and lower cost-effectiveness led NICE as part of the Medical Technologies Evaluation and Diagnostics Assessment Programme [224] to not approve its use as a DPN screening test. We believe this decision is short sighted as the specificity of any test which detects disease earlier is bound to be lower.

### 2.11. Laser Doppler Flare

Cholinergic C-fibres can be activated by heating or iontophoresis of acetylcholine or histamine to induce local vasodilation [149,176,225,226,227,228,229,230,231,232,233,234,235] and the subsequent neurogenic flare can be measured using Laser Doppler flowmetry (LDF) or Laser Doppler imaging (LDI) to quantify the vasomotor axon reflex [225]. LDIFlare is significantly reduced in participants with IGT and T1D compared to controls (2.78 ± 1.1 cm^2^ vs. 5.23 ± 1.7 cm^2^; *p* < 0.0001 and 5.16 ± 2.3 cm^2^ vs. 5.23 ± 1.7 cm^2^; *p* = 0.002) [234], but does not differ between painful and painless DPN nor between participants with or without ulcers [192,236].

An LDIFlare threshold of 3.66 cm^2^ yielded a sensitivity of 75%, specificity of 85%, PPV of 74% and an NPV of 86% for the identification of DPN [230]. An age-dependent reduction of LDIFlare size of 0.56 cm^2^ per decade has been identified [231] and age-specific disease threshold values applied to the LDIFlare threshold of 3.66 cm^2^ yielded a sensitivity of 77%, specificity 90%, PPV of 82% and an NPV of 87% [230]. LDIFlare has demonstrated moderate to high sensitivity and specificity for the detection of DPN, with excellent correlation between the left and right foot (*r* = 0.95, *p* < 0.0001) [231]. At present, LDI does not have a standardised method of analysis and limited normative data and disease threshold values [225,234].

### 2.12. Corneal Confocal Microscopy

Over the past two decades, evaluating corneal nerve morphology has become established as a surrogate marker for diabetic and other peripheral neuropathies [237]. Corneal Confocal Microscopy (CCM) is a rapid, non-invasive, re-iterative ophthalmic imaging modality which visualises small nerve fibre in the corneal sub-basal plexus [238]. Examples of CCM images from a healthy control participant, a participant with diabetes and a participant with DPN are shown in Figure 1. CCM has demonstrated small fibre degeneration in a range of neuropathies including HIV neuropathy, idiopathic small fibre neuropathy, hereditary sensory motor neuropathy and chemotherapy-induced peripheral neuropathy [239,240,241,242,243,244,245]. Standardised corneal nerve morphometric parameters include corneal nerve fibre length (CNFL), fibre density (CNFD), branch density (CNBD), and inferior-whorl length (IWL). CCM has a high sensitivity (60–91%) and specificity (40–87%) for the diagnosis of DPN (Table A5). Decreased corneal nerve parameters have been reported in people with impaired glucose tolerance [246,247], T1D [248,249,250,251,252,253] and T2D [254,255,256,257]. Moreover, reductions in corneal nerve fibres occur in patients with clinically confirmed DPN compared to those without DPN and correlate with DPN severity [258,259,260,261,262,263,264]. The Addition-Denmark study of participants with T2D found no difference in CNFD between participants with and without DPN [86], however the corneal nerve analysis was undertaken using automated analysis and both groups had good and comparable metabolic control [86]. A recently published study identified corneal nerve loss in participants with T1D and T2D compared to controls with good diagnostic accuracy for participants with DPN [265].

A reduction in corneal nerve measures may precede reduced corneal sensitivity in patients with diabetes [266]. In a large study of 590 patients with diabetes, rapid loss of ≥6% of CNFL per year occurred in 17% of participants and was associated with the development of DPN [267]. Thus, a rapid decline in corneal nerves may help to stratify patients at highest risk for the development and progression of DPN. A rapid decline in CNFL, CNFD and CNBD preceded the development of foot ulceration and Charcot foot whilst VPT and QST remained unchanged, suggesting CCM could identify high-risk patients [258,268]. CCM can detect early nerve regeneration with no change in other measures of neuropathy in people with T1D after simultaneous pancreas and kidney transplantation [14,16,269].

Normative reference values have been reported from 343 healthy volunteers with a linear age-dependent decrease in CNFL (−0.045 mm/mm^2^; *p* = 0.07 and −0.060 mm/mm^2^; *p* = 0.02) and CNFD (−0.164 no/mm^2^; *p* < 0.01 and −0.161 no/mm^2^; *p* < 0.01) in both men and women [270]. A pooled multi-national consortium study of 998 participants with T1D (*n* = 516) and T2D (*n* = 482) has demonstrated that a disease threshold of <8.6 mm/mm^2^ can be used to diagnose DPN, whilst a threshold ≥15.3 mm/mm^2^ is sufficient to exclude DPN using fully automated analysis with an equal error rate of 88% specificity and 88% sensitivity [271]. Automated analysis is much quicker than manual human annotation with comparable performance for the detection of DPN [261]. Notably, automated analysis of nerve fibres in CCM images is capable of distinguishing between patients with and without DPN minimising inter/intra observer variability [272,273,274]. The application of a deep learning algorithm on CCM images from 90 healthy participants and 132 people with DPN had a specificity of 87% and sensitivity of 68% for the identification of DPN [275]. 

CCM is widely used in specialist ophthalmology centres worldwide and has a growing user base in neurology which has increased from an initial 3 centres to over 100 centres worldwide within the last two decades. CCM maytherefore be deployed in a screening programme alongside diabetic retinal screening to monitor the development and progression of DPN [276]. Future upscaling in the general clinical (non-research) arena requires the mitigation of barriers for adoption into clinical practice. General clinician acceptance and cost-effectiveness models are still required to tackle downstream barriers. A future plan for clinical adoption will need to tackle both internal (consumer awareness and engagement with CCM manufacturers) and internal factors (structures to encourage adoption, trials/studies of efficacy and policy). A robust framework for generating evidence for deploying this diagnostic methodology will require adoption [2]. However, given the rapid expansion of CCM as a tool for diabetic neuropathy research, additionally with recent clinical trials utilising it as a surrogate endpoint [1], there is a clear expectation for the continued increased use in CCM.

### 2.13. Limitations of This Review

The primary limitation of this critical appraisal is that no formal tool for the assessment of bias or methodological quality was performed. Therefore, the appraisal of the included articles is limited to the authors’ opinions without the utilisation of a validated critical appraisal tool. Additionally, a peer reviewed search strategy and protocol were not published prior to the submission of this manuscript. Significant heterogeneity existed between the studies including different clinical settings, screening tools and diagnostic methods as comparators for the detection of DPN, thus resulting in a challenging comparative interpretation of the diagnostic efficacy of each method. Notably, the majority of articles include participants with a longer duration of diabetes. Any future systematic reviews of the diagnostic efficacy should formally account for study heterogeneity, e.g., study methodology, participant’s characteristics, and definition of DPN.

## 3. Discussion

The identification of early DPN allows for proactive multi-factorial intervention to limit progression of nerve damage. Although guidance exists for the use of a range of simple diagnostic modalities [6,78], current screening techniques detect advanced disease, where interventions are ineffective as summarized in Figure 2. NCS together with signs and symptoms are not sensitive for identifying subclinical DPN. QST is subjective and cannot differentiate between central and peripheral nerve damage. The DNFS static QST protocol identifies patterns of small fibre deficits and may be key to assessing optimal therapeutic response. However, the application of the full battery of tests is time-consuming and requires training. LDIFlare is non-invasive, relatively fast and is associated with a high sensitivity and specificity for DPN. However, there are no published disease threshold values and no standardized method of image analysis. Skin biopsy is considered to be the reference standard for the identification of small fibre neuropathy. However, mass screening and repeat biopsies are not feasible. LEPs, CHEPs, Sudoscan and Neuropad are alternatives but lack robust data regarding diagnostic and prognostic ability. The use of screening tests to identify small fibre pathology in people with diabetes alongside retinopathy screening in a one stop microvascular screening appointment was recently implemented enabling new diagnoses of DPN, which was valued by patients [277].

As such, a new methodology to detect early changes in DPN should be implemented at key stages of patient assessment as shown in Figure 3. CCM provides detailed quantification of small nerve fibres and predicts the development and progression of DPN. The anatomically distinct area of examination (cornea) from the perceived primary area of pathology (feet and hands) have been debated as a possible limitation of CCM. However, as the need for corneal transparency decrees the lack of vasculature, and unlike unmyelinated fibres elsewhere, corneal nerve fibres have a significant vulnerability to degeneration from metabolic, toxic, immune or inflammatory insults [1]. Corneal nerve fibre pathology is demonstrated in a number of systemic and central neurodegenerative diseases, e.g., Parkinson’s, multiple sclerosis, stroke [3,4]; this is suggestive that structural changes in corneal nerves are not entirely specific to the peripheral nervous system [5]. Clearly, the corneal sub-basal nerve plexus is vulnerable to a number of disease mechanisms and corneal nerve pathology may manifest in diverse conditions. Whereas, DPN is characterised by a distal and symmetric degeneration of sensory nerves. CCM conveniently images exactly these most distal sensory nerves of the cornea. There is now a robust evidence base for the role of CCM in DPN. Importantly, recent studies identifying greater corneal pathology of the inferior whorl (a vertex of nerves located infero-nasally) may discriminate between pDPN and insensate DPN [6,7]. The prognostic role of the inferior whorl in those with lower inferior whorl nerve length in the development of neuropathic pain requires further investigation. The early detection of DPN is essential in the prevention of long-term sequelae and in reducing morbidity. Early detection of DPN requires the assessment of small nerve fibres, which are of paramount importance. CCM is rapid, non-invasive and readily repeatable, providing objective, reproducible and sensitive quantification of small nerve fibres and the detection of nerve degeneration and regeneration. CCM is a game changer in the diagnosis and evaluation of DPN and represents the most pertinent modality for early detection.

## Figures and Tables

**Figure 1 diagnostics-11-00165-f001:**
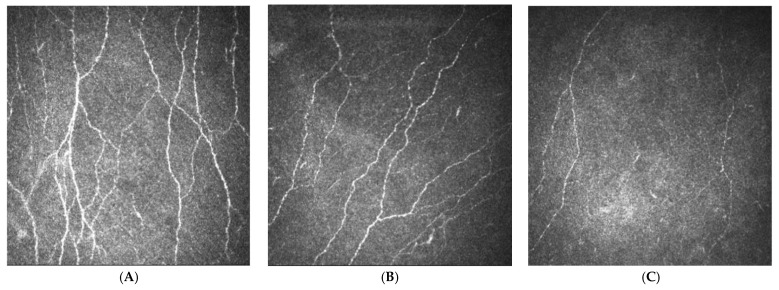
Images taken using corneal confocal microscopy in a healthy control participant (**A**), a participant with diabetes (**B**) and a participant with severe diabetic peripheral neuropathy (**C**) demonstrating the progressive corneal nerve fibre loss in the Bowman’s layer.

**Figure 2 diagnostics-11-00165-f002:**
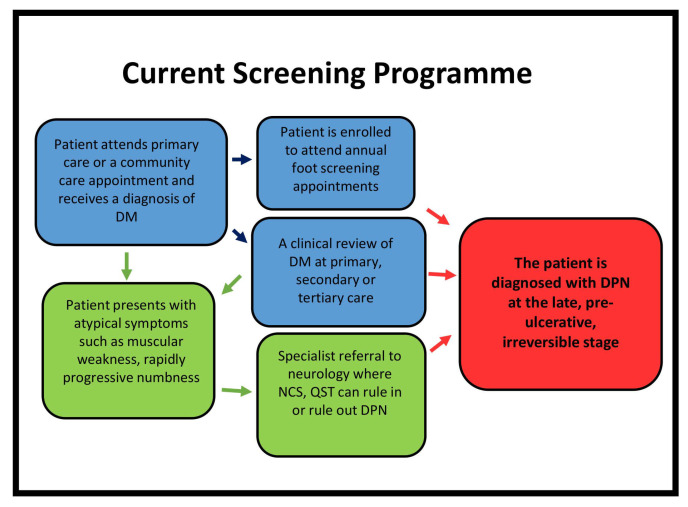
Clinical pathway for patients diagnosed with diabetes mellitus. DM—diabetes mellitus; DPN—diabetic peripheral neuropathy; NCS—nerve conduction studies; QST—quantitative sensory testing.

**Figure 3 diagnostics-11-00165-f003:**
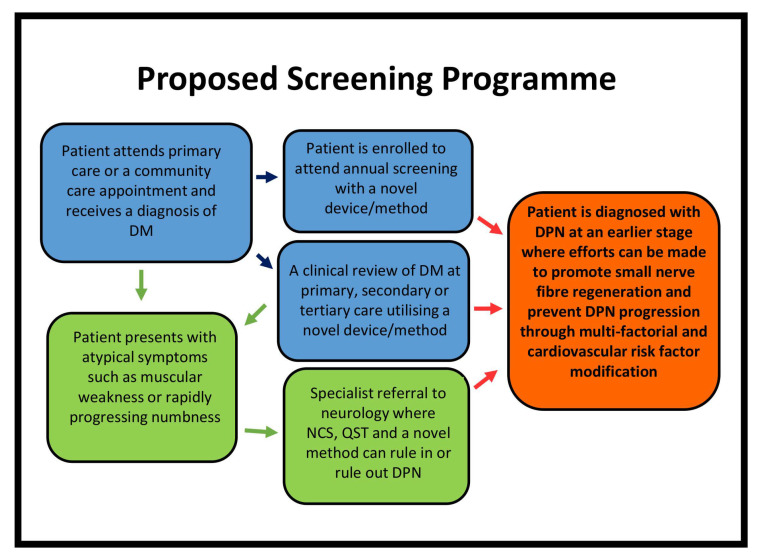
Proposed clinical pathway for patients diagnosed with diabetes mellitus using a new screening method. DM—diabetes mellitus; DPN—diabetic peripheral neuropathy; NCS—nerve conduction studies; QST—quantitative sensory.

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
