# Peer review of "Early Detection of Diabetic Peripheral Neuropathy: A Focus on Small Nerve Fibres"

_diagnostics, 2021, doi:10.3390/diagnostics11020165_

Round 1
Reviewer 1 Report
The review titled “Early Detection of Diabetic Peripheral Neuropathy: A Focus on Small Nerve Fibers” appraised research and clinical methods for the diagnosis or screening of early DPN and indicated that improved screening tools will help reduce and prevent foot ulceration and amputations and reduce the frequency of this common microvascular complication. This is a very mature manuscript and easy to follow. The screening pathway was clearly shown in figures and help the reader to understand the importance of early diagnosis. All the screening and diagnostic tools are well introduced in the article. The author also organized the sensitivity and specificity for DPN diagnosis reported by different literature. The only suggestion I have is the discussion part is too short and didn’t give much direction on what we need to improve in the future. Although the author mentioned that CCM is rapid, non-invasive, and can be repeated as often as desired… What is the weakness of this method, what we need to improve, why it’s not widely used should be discussed.
Author Response
We thank the reviewers and associate editor in reviewing our manuscript. Thank-you for your review of our submitted manuscript. Please see our point-by-point response.
Q1.) Although the author mentioned that CCM is rapid, non-invasive, and can be repeated as often as desired… What is the weakness of this method, what we need to improve, why it’s not widely used should be discussed.
A1.) Thank you for the opportunity to respond to this question. This is partly addressed in the second and third items of our rebuttal however the following paragraph has been added to the Corneal Confocal Microscopy section (Lines 462-470, page 10):
“CCM is widely used in specialist ophthalmology centres worldwide and has a growing user base in neurology which has increased from an initial 3 centres to over 100 centres worldwide. CCM could therefore be deployed in a screening programme alongside diabetic retinal screening to monitoring the development and progression of DPN [283]. For future upscaling in the general clinical (non-research) arena requires mitigation of barriers for adoption into clinical practice. General clinician acceptance and cost-effectiveness models are still required to tackle downstream barriers.“
The following has also been added to address this comment into the discussion (lines 517-521, page 11)
“The anatomically distinct area of examination (cornea) from the perceived primary area of pathology (feet and hands) have been debated as a possible limitation of CCM. However, as the need for corneal transparency decrees the lack of vasculature, and unlike unmyelinated fibres elsewhere, corneal nerve fibres have a significant vulnerability to degeneration from metabolic, toxic, immune or inflammatory insults [1].”
Further, a key challenge for the adoption of CCM in clinical centres is that Heidelberg Engineering is the sole manufacturer at the moment. Users would benefit from a more diversified supply chain e.g. versatility, specifications and initial investment. We have inserted the following paragraph into the Corneal Confocal Microscopy section (Lines 470-476, page 10) regarding the adoption of this technology in relation to CCM manufacturers.
“A future plan for clinical adoption will need tackle both internal (consumer awareness and engagement with CCM manufactures) and internal factors (structures to encourage adoption, trials/studies of efficacy and policy). A robust framework for generating evidence for developing this novel diagnostic methodology will require adoption [2]. However, given the rapid expansion of CCM as a tool for diabetic neuropathy research, additionally with recent clinical trials utilising it as a surrogate endpoint [1], there is a clear expectation for the continued increase in CCM use.”
Q2.) The only suggestion I have is the discussion part is too short and didn’t give much direction on what we need to improve in the future.
A1. Thank you for your feedback from our discussion in the manuscript. We have added the following paragraph (Lines 512-535, pages 11-12) into the discussion for your consideration:
“Corneal nerve fibre pathology is demonstrated in a number of systemic and central neurodegenerative diseases e.g. Parkinson’s, multiple sclerosis, stroke [3,4]; this is suggestive that structural changes in corneal nerves are not entirely specific to the peripheral nervous system [5]. Clearly, the corneal sub‐basal nerve plexus is vulnerable to a number of disease mechanisms and corneal nerve pathology may manifest in diverse conditions. Whereas, DPN is characterised by a distal and symmetric degeneration of sensory nerves. CCM conveniently images exactly these most distal sensory nerves of the cornea. There is now a robust evidence base for the role of CCM in DPN. Importantly, recent studies identifying greater corneal pathology of the inferior whorl (a vertex of nerves located infero-nasally) may discriminate between pDPN and insensate DPN [6,7]. The prognostic role of the inferior whorl in those with lower inferior whorl nerve length in the development of neuropathic pain requires further investigation. The early detection of DPN is essential in the prevention of long-term sequelae and in reducing morbidity. Early detection of DPN requires the assessment of small nerve fibres, which are of paramount importance. CCM is rapid, non‐invasive and readily repeatable, providing objective, reproducible and sensitive quantification of small nerve fibres and the detection of nerve degeneration and regeneration. CCM is a game changer in the diagnosis and evaluation of DPN and represents the most pertinent modality for early detection."
Reviewer 2 Report
This is a very interesting assessment of screening and diagnosis of a significant complication of Diabetes that affect millions around the globe. The focus of the review is to argue for the need of early detection of Diabetic Peripheral Neuropathy (DPN). The review in my opinion was extensive but the authors did not elaborate on how the studies were identified. While, rightly, they do not claim that this is a systematic review they should at least explain how this manuscript was conceptualised and what was the main aim and their methodology in reviewing this extensive body of literature. They may do this early in the introduction if they do not want a method section. There is only one statement in the abstract that this is a critical appraisal of the studies and clinical practice of screening and diagnosis of DPN. If it is a critical appraisal, then the authors must explain more on how the studies were assessed and what was the approach in detecting bias…etc
The second point I would like to raise is that there is less coverage of painful DPN and their screening tools such as DN4 and LANSS. I understand that in many cases the pDPN will appear late at the course of condition and may be absent but at least authors should alert to this.
I have enjoyed reading this good manuscript, but I think it may be categorized better as a discussion article or an opinion piece and acknowledge the limitations of this review.
Author Response
We thank the reviewers and associate editor in reviewing our manuscript. Thank-you for your review of our submitted manuscript. Please see our point-by-point response.
Q1.) The review in my opinion was extensive but the authors did not elaborate on how the studies were identified. While, rightly, they do not claim that this is a systematic review they should at least explain how this manuscript was conceptualised and what was the main aim and their methodology in reviewing this extensive body of literature. They may do this early in the introduction if they do not want a method section. There is only one statement in the abstract that this is a critical appraisal of the studies and clinical practice of screening and diagnosis of DPN. If it is a critical appraisal, then the authors must explain more on how the studies were assessed and what was the approach in detecting bias…etc. .) I have enjoyed reading this good manuscript, but I think it may be categorized better as a discussion article or an opinion piece and acknowledge the limitations of this review.
Thank you for highlighting this oversight. We agree our manuscript appraised studies through a narrative review of literature, primarily comprising of the authors’ opinions rather than a formal validated critical appraisal tool. Our intention is to create discussion within the field of DPN. We have added a short paragraph detailing the study selection process (Lines 132-140, pages 3-4):
“1.5. Methods
Electronic database searches were undertaken in Google Scholar, EMBASE, PubMed, OVID and Cochrane CENTRAL to identify the included articles. Reference lists of relevant articles were searched and in addition, studies were identified by authors with expertise in DPN. Studies published from initial curation of the electronic database to November 2020 were identified and those felt not relevant by authors were excluded with the guidance of the senior author (U.A.).”
We have also added a limitations section before the discussion “2.13. Limitations of the review” (Lines 478-489, pages 10-11):
"2.13. Limitations of this review
The primary limitation of this critical appraisal is that no formal tool for the assessment of bias or methodological quality was performed. Therefore, the appraisal of the included articles is limited to the authors’ opinions without the utilisation of a validated critical appraisal tool. Additionally, a peer reviewed search strategy and protocol were not published prior to the submission of this manuscript. Significant heterogeneity existed between the studies including different clinical settings, screening tools and diagnostic methods as comparators for the detection of DPN, thus resulting in a challenging comparative interpretation of the diagnostic efficacy of each method. Notably, the majority of articles include participants with a longer duration of diabetes. Any future systematic reviews of the diagnostic efficacy should account for study heterogeneity e.g. study methodology, participant’s characteristics, and definition of DPN."
Q2.) The second point I would like to raise is that there is less coverage of painful DPN and their screening tools such as DN4 and LANSS. I understand that in many cases the pDPN will appear late at the course of condition and may be absent but at least authors should alert to this.
A2.) Thank you for identifying this oversight. We agree that this is an important aspect of screening as neuropathic pain can be an early sign of neuropathy in pre-diabetes, diabetes and metabolic syndrome. We have included the following paragraph (lines 145-155, page 4):
“Confirmed DPN represents a subtle and gradual disease process, in which the symptoms are often unreliable as an indicator of early nerve damage [80,81]. However, neuropathic pain may be the initial presenting symptom of diabetic neuropathy in patients with diabetes, pre-diabetes or metabolic syndrome [8]. Thus, validated screening instruments which utilise sensory and affective verbal pain descriptors (burning sensation, tingling/prickling, numbness, electric shocks, pain evoked by light touch) such as the Leeds Assessment of Neuropathic Symptoms and Signs (LANSS), douleur neuropathique en 4 (DN4) and painDETECT are widely used for the identification of neuropathic pain in diabetes [9-14]. Notably, Bennet et al [15] demonstrated the utility of screening for neuropathic pain in a population of people with diabetes using a postal self-completed LANSS questionnaire, highlighting the excess prevalence and burden of pDPN. However, measures such as the neuropathy symptom score (NSS) cannot reliably identify early DPN [82,83].”
We have also included an additional article in Table B5 “Studies of the sensitivity and specificity of corneal confocal microscopy for the diagnosis of DPN” (Pages 29-30) which has recently been published.